# Proposal for Customer Identification Service Model Based on Distributed Ledger Technology to Transfer Virtual Assets

**Keundug Park [1]** and **Heung-Youl Youm [2],***

1    AI&Blockchain Research Center, Seoul University of Foreign Studies, Seoul 60745, Korea; jacepark926@gmail.com
2    Department of Information Security Engineering, Soonchunhyang University, Asan 31538, Korea
*    Correspondence: hyyoum@sch.ac.kr

**Abstract:** Recently, cross-border transfers using blockchain-based virtual assets (cryptocurrency) have been increasing. However, due to the anonymity of blockchain, there is a problem related to money laundering because the virtual asset service providers cannot identify the originators and the beneficiaries. In addition, the international anti-money-laundering organization (the Financial Action Task Force, FATF) has placed anti-money-laundering obligations on virtual asset service providers through anti-money-laundering guidance for virtual assets issued in June 2019. This paper proposes a customer identification service model based on distributed ledger technology (DLT) that enables virtual asset service providers to verify the identity of the originators and beneficiaries.

**Keywords:** blockchain; distributed ledger technology; cryptocurrency; virtual asset; customer identification; customer due diligence; travel rule

## 1. Introduction

Cross-border transfers using blockchain-based virtual assets (VAs) are rising every year. The global fiat currency transfer market is estimated at USD 1.9 trillion as of 2017 [1], and the amount of cross-border transfers in Korea is estimated at USD 13.4 billion as of 2018 [2]. With the development of blockchain and distributed ledger technology (DLT), the virtual asset transfer market is expected to replace the fiat currency transfer market with its fast transfer time and low fees. Although the fee rate is different in each country, the average fee rate for a cross-border transfer based on USD 200 as of 2017 is 7.14% (approximately USD 14), and in the case of virtual asset transfer, a lower fee rate can be applied to reduce the burden on customers [3].

However, due to the anonymity of the blockchain, there is a problem related to money laundering because the virtual asset service provider (VASP) cannot identify the originator and the beneficiary. In addition, the international anti-money-laundering organization (the Financial Action Task Force, FATF) has placed anti-money-laundering obligations on virtual asset service providers through anti-money-laundering guidance for virtual assets issued in June 2019.

This paper proposes a customer identification service model based on DLT that enables virtual asset service providers to verify the identity of the originators and the beneficiaries so that virtual asset service providers can implement anti-money-laundering (AML) in accordance with customer due diligence (CDD).

The contribution of this paper is as follows: to our best knowledge there is no previous work that has studied a customer identification system that fully meets all requirements defined by the FATF, our proposed service model provides a framework to share and verify securely customer's minimal identity information among VASPs using a DLT system that incorporates VASPs and IDSPs.

This paper is organized as follows. Section 1 introduces virtual asset transfer market trends and AML, Section 2 describes virtual asset transfer and solution to customer identi-

fication, Section 3 describes related studies including a problem on customer identification under the existing blockchain and DLT environment for virtual asset transfers, Section 4 proposes a customer identification service model based on DLT for virtual asset transfers to solve the problem identified in Section 3, Section 5 identifies security threats to the proposed service model and provides security requirements against the security threats, and Section 6 discusses the results and briefly concludes.

## 2. Background

This section describes virtual asset transfer under the existing blockchain and DLT platform and a solution to customer identification.

### 2.1. Virtual Asset Transfer

In the existing blockchain and DLT platform environment, virtual asset transfer means transferring ownership of the virtual asset held by the originator to the beneficiary. This is called a transaction. For example, in the case of Ethereum transfer, the information required for the transaction between the originator and beneficiary is the originator's digital wallet address, the beneficiary's digital wallet address, and the amount of Ether [4].

In Figure 1, the "From" means the originator's digital wallet address, the "To" means the beneficiary's digital wallet address, and the "Value" means the amount of Ether. However, the identity of the originator and beneficiary cannot be verified. This anonymity has been misused for money laundering and terrorism financing using virtual assets.

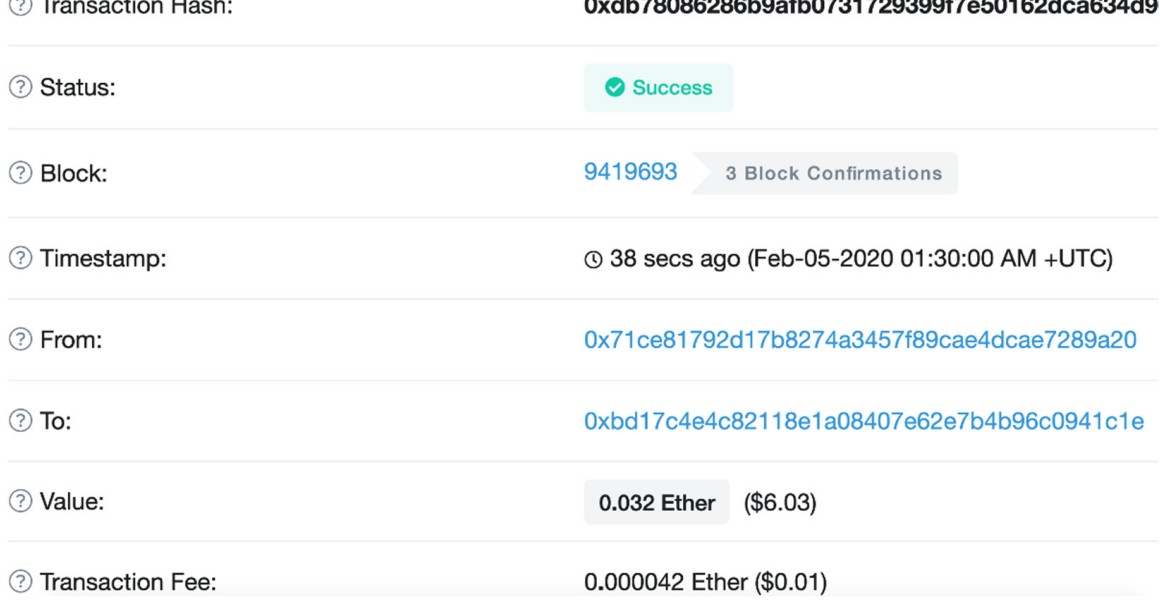

**Figure 1.** Example of details on Ethereum transfer [4].

### 2.2. Solution to Customer Identification

Many virtual asset service providers (VASPs) verify their customers' identities, but it is very difficult to share customer's identity information between VASPs. In other words, when a cross-border virtual asset (VA) is transferred, the originator's VASP cannot verify the beneficiary's identity information, and also the beneficiary's VASP cannot verify the originator's identity information.

This paper proposes a service model in which the VASPs can verify and share the identity of the originator and beneficiary prior to transferring virtual assets. In Figure 2, the VASP-1 verifies the originator's identity and securely shares the originator's identity information to other VASPs and verifies beneficiary's identity information. The VASP-2 verifies the beneficiary's identity and securely shares the beneficiary's identity information

to other VASPs and verifies the originator's identity information. The proposed system connects the VASPs using DLT and operates totally independently from the existing blockchain platform (e.g., Ethereum, etc.) which transfers VAs. The VASP-1 sends the originator's personal data to VASP-2 with existing standard protocols. The VASP-2 sends the beneficiary's personal data to VASP-1 with existing standard protocols (e.g., SAML [5], OpenID Connect [6], and TLS [7,8]).

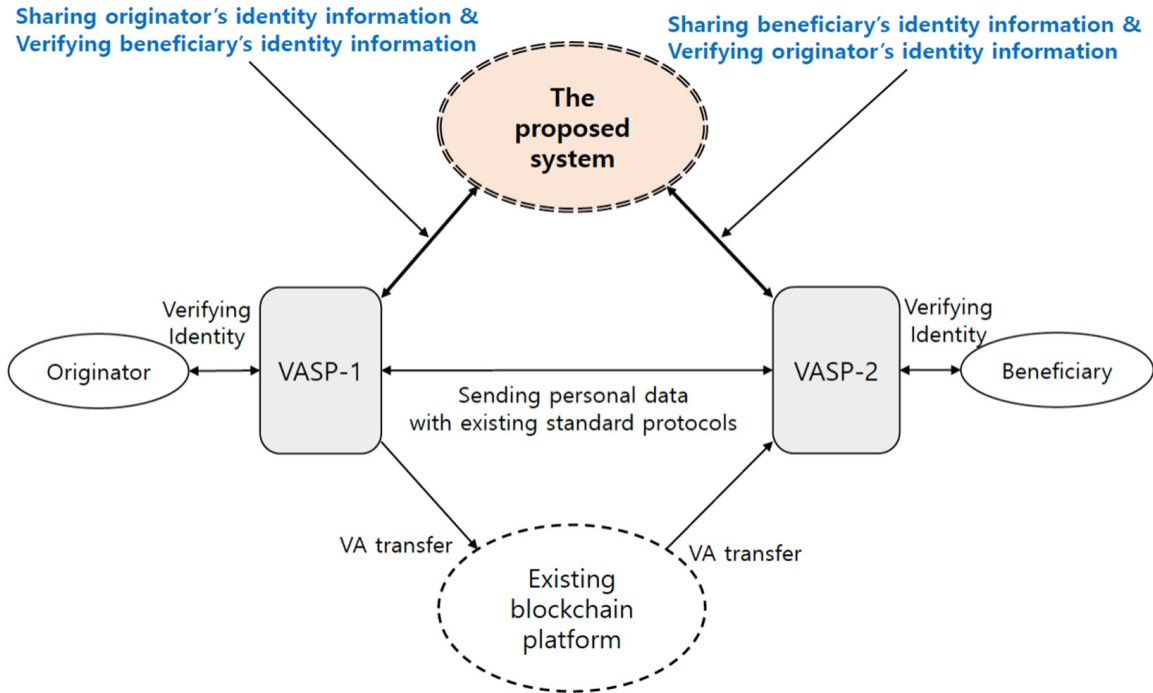

**Figure 2.** Service model for customer identification.

## 3. Related Studies

This section describes terms and definitions, regulation issues regarding AML and privacy, a problem with customer identification under the existing blockchain and DLT platforms, and several works from others.

### 3.1. Terms and Definition

- A virtual asset (VA) is a digital representation of value that can be digitally traded, or transferred, and can be used for payment or investment purposes [9];
- Virtual asset service provider (VASP) means any natural or legal person that, as a business, conducts one or more of the activities or operations (e.g., exchange, transfer, safekeeping, etc.) for or on behalf of another natural or legal person [9];
- Off-chain: related to a blockchain system, but located, performed, or run outside that blockchain system [10];
- On-chain: located, performed, or run inside a blockchain system [10];
- Identity service provider (IDSP): a generic umbrella term that refers to all of the various types of entities involved in providing and operating the processes and components of a digital ID system or solution. IDSPs provide digital ID solutions to users and relying parties [11];
- Permissioned distributed ledger system: distributed ledger system in which permissions are required to maintain and operate a node [10];
- Private distributed ledger system: distributed ledger system that is accessible for use only to a limited group of DLT users [10];
- Know your customer (KYC): process to verify the identity of a customer in order to prevent financial crime, money laundering, and terrorism financing;

- Customer due diligence (CDD): a process where relevant information about the customer is collected and evaluated for any potential risk for the organization or money laundering/terrorist financing activities.

### 3.2. Regulation Issues of CDD for AML

The FATF currently comprises 37 member jurisdictions and two regional organizations, representing most major financial centers in all parts of the globe [12].

The FATF published a guidance named "GUIDANCE FOR A RISK-BASED AP-PROACH TO VIRTUAL ASSETS AND VIRTUAL ASSET SERVICE PROVIDERS" and updated a Recommendation named "INTERNATIONAL STANDARDS ON COMBATING MONEY LAUNDERING AND THE FINANCING OF TERRORISM & PROLIFERATION" in June 2019. Both Recommendation 15—New Technologies and Interpretative Note to Recommendation 15 (INR. 15) in the FATF Guidance are important for virtual asset service providers. In particular, Section 7(a) and (b) in the INR. 15 presents CDD subjected to VASPs as follows:

> 7. (a) R.10—The occasional transactions designated threshold above which VASPs are required to conduct CDD is USD/EUR 1000;

> 7. (b) R.16—Countries should ensure that originating VASPs obtain and hold required and accurate originator information and required beneficiary information on virtual asset transfers. Countries should ensure that beneficiary VASPs obtain and hold required originator information and required and accurate beneficiary information on virtual asset transfers. The same obligations apply to financial institutions when sending or receiving virtual asset transfers on behalf of a customer. [9,13,14]

VASPs could conduct CDD using digital identity from their customer. The reason is that, according to guidance on digital identity published by the FATF, Section III (FATF STANDARDS ON CUSTOMER DUE DILIGENCE), Article 79 presents the following:

> Article 79 Recommendation 10 is technology-neutral. Recommendation 10 (a) permits financial institutions to use "documents" as well as "information or data", when conducting customer identification and verification. Recommendation 10 (a) does not impose any restrictions on the form (documentary/physical or digital) that identity evidence—"source documents, information or data"—can take. [11]

The Interpretative Note to Recommendation 16 (INR. 16) in FATF Recommendation presents information accompanying cross-border transfers. Information accompanying all qualifying wire transfers should always contain the name of the originator; the originator account number; the originator's address, national identity number, customer identification number, or date and place of birth; the name of the beneficiary; and the beneficiary account number [15].

### 3.3. Regulation Issues of Privacy

The EU general data protection regulation (GDPR) enacts the pseudonymisation of personal data and the right to be forgotten such as Article 4 Definitions and Article 17 Right to erasure [16].

The administrative safety authority in Korea has requested the National Assembly to partially amend privacy-related law. As of January 2020, the law was processed by the National Assembly and transferred to the government. The partial amendment law is presented briefly as follows:

- Article 2: Definitions (1–2): "A pseudonymization" means the removal of a part of personal data, or the replacement of part or all, such that a particular individual is not identifiable without additional information;

- Article 21: Destruction of personal data (1): The personal data processor shall destroy the personal data without delay when the data becomes unnecessary, such as the expired retention period or the achievement of the purpose of processing the data;
- Article 28-2: Processing pseudonym information (1): The personal data processor may process the pseudonym information without the consent of the data subject for statistical preparation, scientific research, and public record preservation [17].

### 3.4. Problem with Customer Identification

As mentioned in Section 3.2, although VASPs should comply with CDD for AML, there is a problem with customer identification when transferring virtual assets under the existing blockchain and DLT platform environment. The problem is due to the anonymity of the existing blockchain and DLT platforms. According to Hyperledger Fabric and Ethereum, which are a well-known and widely used DLT platform, they do not verify the identity of the owner when creating a digital wallet and do not verify the identity of the originator and the beneficiary when storing a transaction record for the transfer of virtual assets [18,19]. The anonymity of digital wallet owners increases the number of cases in which virtual assets are misused for money laundering and terrorism financing [20]. In the financial sector, the anonymity is also identified as a potential security threat [21].

In Figure 3, if the originator transfers virtual assets to the beneficiary under the existing blockchain and DLT platform, the VASP-1 can verify the identity of the originator as a customer and the VASP-3 can verify the identity of the beneficiary as a customer. However, it is difficult for the VASP-1 to verify and obtain identity information of the beneficiary from the VASP-3, and it is also difficult for the VASP-3 to verify and obtain identity information of the originator from the VASP-1.

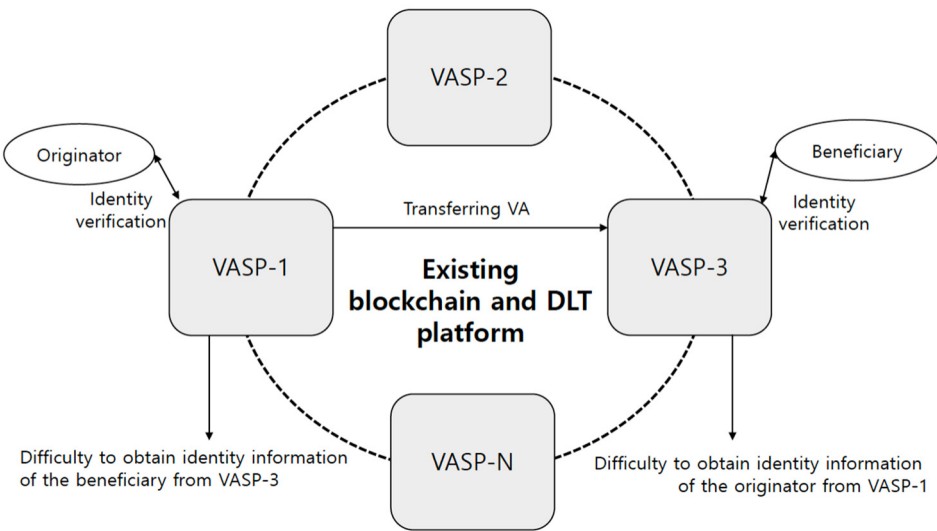

**Figure 3.** Anonymity regarding virtual asset transfers.

In particular, it is practically impossible for VASPs to verify the identity of the originator and the beneficiary when cross-border virtual asset transfers. There is no infrastructure to share customer identity information among VASPs.

### 3.5. Other Approaches for Customer Identification

Several organizations and papers have proposed ways to solve the problem mentioned in Section 3.4 as follows, but their proposals differ from the proposed service model in terms of concept and concreteness.

According to the VASPnet, communication between virtual asset service providers (VASPs) and between VASPs and financial institutions (FI), and between FIs relating to virtual assets has been proposed as a new area for standardization [22]. The white paper of

the OpenVASP outlines an open protocol among VASPs for the mutual exchange of originator and beneficiary information. It does so in a fully decentralized manner, leveraging cryptographically secure peer-to-peer communication and capabilities of the Ethereum blockchain for authentication. The protocol works with any blockchain or distributed ledger technology (DLT) used for the underlying virtual asset transfer. It puts the privacy of transferred data at the center of its design [23]. A peer-to-peer mechanism to comply with the FATF regulations with minimal cost impact to participants and consideration for preserving high-performance transaction processing at cryptocurrency virtual asset service providers are proposed in [24]. The Shyft is an Ethereum blockchain-based protocol that enables the secure and auditable sending of messages between individual users and trusted parties. The Shyft leverages the participation of these parties and their ability to onboard users in accordance with existing compliance while adding the ability to broadcast attestations of relevant information on user data to other parties by request, assuming user consent is present [25]. The Sygna Bridge is a simple API-based messaging service that enables VASPs to privately share all required transmittal compliance data [26].

A new system based on distributed ledger technology (DLT) that reduces the costs of the core KYC (know your customer) verification process for financial institutions and improves customer experience is proposed in [27]. The core KYC verification process is only conducted once for each customer, regardless of the number of financial institutions with which that customer intends to work. The result of the core KYC verification can be securely shared by customers with all the financial institutions that they intend to work with. This paper focuses on the improvement of the KYC verification process at existing banks, not VASP dealing with VA. All banks store customer's personal data and share the result of KYC verification using DLT system.

A privacy-preserving KYC design for Ethereum-based financial services with a third-party KYC provider is proposed in [28]. This paper focuses on verifying customer identity with third-party KYC provider storing their personal data when exchanging tokens based on Ethereum. On the contrary, our proposed service model can be applied to any type of VA (e.g., token) transfer because the DLT security platform consisting of ON-CHAIN and OFF-CHAIN is independent of any blockchain (e.g., Ethereum, etc.) platforms.

A VASP claims exchange network in which VASPs can deliver signed claims (obtained from their Claims Providers) about subjects (i.e., customers) and public-key information or certificates (i.e., key ownership information) to other VASPs in a secure and confidential manner is proposed in [29]. This paper focuses on the concept and requirements of the claim exchange networks for VA transfers, which consist of the customer (i.e., originator, beneficiary), certificate authority (CA), VASPs, claims providers, and data providers. In particular, it has partially underpinned the technical background of our proposed service model with concept of decentralized claims exchange networks.

The notion of a trust network of VASPs in which originator and beneficiary information, including key ownership information that can be exchanged securely while observing individual privacy requirements, is proposed in [30]. This paper focuses on the concept and requirements of the trust network of VASPs sharing customer certificates and attributes and public key certificate management for VA transfers. In particular, it has partially underpinned the technical background of our proposed service model with the concept of trust network of VASPs.

A framework for regulating cryptocurrency payments intermediaries based on Article 4A of the Uniform Commercial Code in the United States is proposed in [31]. This paper focuses on topics for inclusion in a commercial law for cryptocurrency payments intermediaries that are found in other payments and e-payments commercial or consumer protection laws.

The key potential risks, legal, and regulatory challenges, possible avenues to address compliance concerns, and technological solutions related to virtual currency (VC) are proposed in [32]. This paper focuses on the necessity to define best practices and standards, as well as training programs, for the next generation of VC.

The objective of this paper is to design a framework to share and verify securely customer's minimal identity information among VASPs using DLT system which incorporates VASPs and IDSPs, where VASP does not store the beneficiary's personal data (e.g., national identity number, etc.) from other VASPs and IDSPs.

## 4. Proposal for Customer Identification Service Model

This section proposes a customer identification service model based on DLT for virtual asset transfers to solve a problem with customer identification mentioned in Section 3.4.

### 4.1. Overview

The DLT-based customer identification service model includes customers, VASPs, IDSPs, DLT security platform, existing blockchain, and DLT platforms, and existing IT service platforms. The DLT security platform meets security functional requirements and pseudonymizes the personal data of customers. The proposed service model provides a framework to securely share and verify customer's minimal identity information among VASPs using DLT system which incorporates VASPs and IDSPs. The IDSPs could use the DID (Decentralized Identifiers) [33] to identify their customer.

In Figure 4, we describe the main components and their roles for customer identification service model based on the DLT security platform. The VASPs and IDSPs participate as nodes of the DLT security platform, in this context referred to as the DLT platform that meets security functional requirements described later in this section. In this diagram, the DLT security platform consists of ON-CHAIN and OFF-CHAIN (see Figure 5) to enable registering, updating, removing, and sharing customer identity information. The VASPs, existing blockchain and DLT platforms, and existing IT service platforms can obtain customer identity information from the ON-CHAIN. The customers can provide identity information to the OFF-CHAIN through either the VASPs or IDSPs.

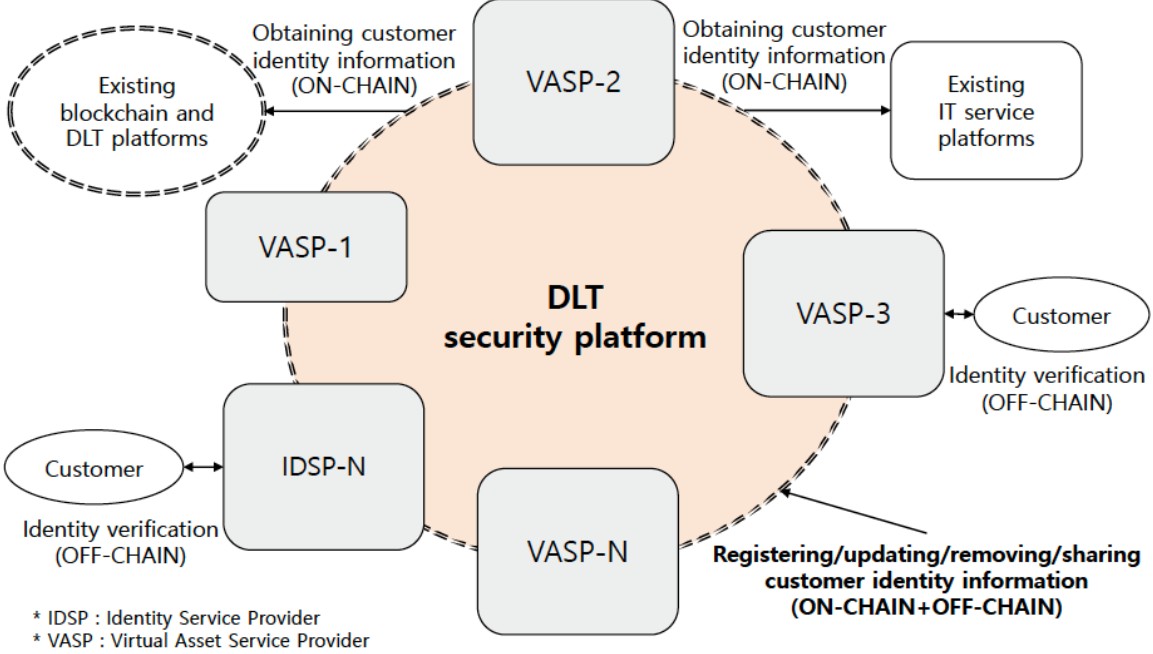

**Figure 4.** A diagram of customer identification service model based on DLT security platform.

In Figure 5, we describe participants as nodes of the ON-CHAIN and OFF-CHAIN and different roles between the ON-CHAIN and OFF-CHAIN. The ON-CHAIN is a kind of repository to deal with customer identity information including personal data de-identified by uni-directional cryptography inside the DLT system. The OFF-CHAIN is a kind of repository to deal with customer identity information including personal data de-identified by bi-directional cryptography outside the DLT system, but it is very closely related to

the ON-CHAIN (see Figure 6). In this diagram, the IDSPs should be participants in the OFF-CHAIN, and the VASPs should be participants in the ON-CHAIN, but the VASPs acting as IDSP can be participants in the OFF-CHAIN. The OFF-CHAIN enables one to register, update, and remove customer identity information de-identified only in the same jurisdiction. The ON-CHAIN enables one to share customer identity information de-identified with VASPs in all countries. A flow chart to determine whether a blockchain is the appropriate technical solution to solve a problem and also differences between permissionless, permissioned blockchains, and a centralized database are proposed in [34]. Both the ON-CHAIN and OFF-CHAIN are the permissioned and private distributed ledger system using consensus mechanism such as practical Byzantine fault tolerance (PBFT), which is identical to the Private Permissioned Blockchain in [34].

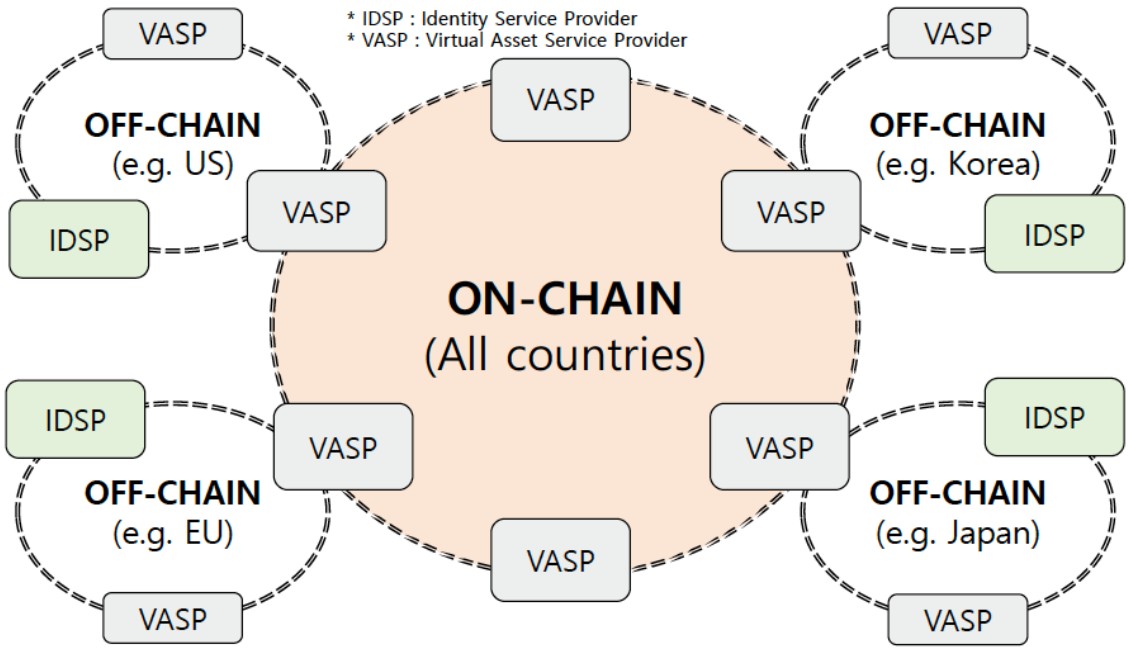

**Figure 5.** A diagram of the DLT security platform consisting of OFF-CHAIN and ON-CHAIN.

The VASPs in many countries need to share their massive customer identity information including personal data before cross-border VA transfers. In addition, it is very important for them to prevent tampering with customer identity information. The proposed service model uses DLT to keep the integrity of customer identity information rather than a centralized database. The proposed service model can be utilized for customer identification service between the VASPs, between the VASP and existing blockchain and DLT service provider, and between the VASP and existing IT service provider, which is required by the FATF [9].

The DLT security platform should satisfy security functional requirements as follows:

- "Identification and Authentication" provides a means of identifying and authenticating users or applications in accordance with identity verification procedures [21]. The solution should meet the following security functional components: FIA_AFL.1 (Authentication failure handling), FIA_ATD.1 (User attribute definition), FIA_SOS.1 (Verification of secrets), FIA_SOS.2 (TSF Generation of secrets), FIA_UAU.2 (User authentication before any action), FIA_UAU.5 (Multiple authentication mechanisms), FIA_UAU.6 (Re-authenticating), FIA_UAU.7 (Protected authentication feedback), FIA_UID.2 (User identification before any action), FTA_MCS.1 (Basic limitation on

multiple concurrent sessions), FTA_SSL.3 (TSF-initiated termination), and FTA_SSL.4 (User-initiated termination) in Common Criteria (CC) [35];

- "Security Audit" provides a means to track the accountability for security incidents by recording the user's behavior logs and keeping it safe [21]. The solution should meet the following security functional components: FAU_ARP.1 (Security alarms), FAU_GEN.1 (Audit data generation), FAU_GEN.2 (User identity association), FAU_SAR.1 (Audit review), FAU_SAR.2 (Restricted audit review), FAU_SAR.3 (Selectable audit review), FAU_SEL.1 (Selective audit), FAU_STG.1 (Protected audit trail storage), FAU_STG.2 (Guarantees of audit data availability), FAU_STG.3 (Action in case of possible audit data loss), FAU_STG.4 (Prevention of audit data loss), and FPT_STM.1 (Reliable time stamps) in Common Criteria (CC) [35];

- "Communication Protection" provides a safe communication means for transmitting critical information between nodes participating in distributed ledger network and provides a safe communication means for transmitting critical information between DLT system and external system [21]. The solution should meet the following security functional components: FCO_NRO.2 (Enforced proof of origin), FCO_NRR.2 (Enforced proof of receipt), FPT_ITC.1 (Inter-TSF confidentiality during transmission), FPT_ITI.1 (Inter-TSF detection of modification), and FTP_ITC.1 (Inter-TSF trusted channel) in Common Criteria (CC) [35];

- "Cryptography Control" provides a means to securely process cryptographic keys for either digital signatures during transaction between users or encryption while transmitting and storing critical data [21]. The solution should meet the following security functional components: FCS_CKM.1 (Cryptographic key generation), FCS_CKM.2 (Cryptographic key distribution), FCS_CKM.3 (Cryptographic key access), FCS_CKM.4 (Cryptographic key destruction), and FCS_COP.1 (Cryptographic operation) in Common Criteria (CC) [35];

- "Access Control" provides a means to control unauthorized access to sensitive information assets [21]. The solution should meet the following security functional components: FDP_ACC.1 (Subset access control) and FDP_ACF.1 (Security-attribute-based access control) in Common Criteria (CC) [35];

- "Privacy Protection" provides a means to securely handle the user's personal data to prevent leakage and exposure [21]. The solution should meet the following security functional components: FPR_PSE.2 (Reversible pseudonymity) and FDP_RIP.2 (Full residual information protection) in Common Criteria (CC) [35];

- "Data Protection" provides a means to securely transmit or store critical information such as user's transaction data and distributed ledger data (including personal data) to prevent leakage or tampering [21]. The solution should meet the following security functional components: FDP_DAU.2 (Data Authentication with Identity of Guarantor), FDP_ETC.1 (Export of user data without security attributes), FDP_ETC.2 (Export of user data with security attributes), FDP_ITC.1 (Import of user data without security attributes), FDP_ITC.2 (Import of user data with security attributes), FDP_RIP.2 (Full residual information protection), FDP_SDI.1 (Stored data integrity monitoring), FDP_SDI.2 (Stored data integrity monitoring and action), FDP_UCT.1 (Basic data exchange confidentiality), and FDP_UIT.1 (Data exchange integrity) in Common Criteria (CC) [35];

- "Resource Availability" provides a means to maximize system availability in response to lack of system resources (e.g., CPU, memory, network, and storage) of nodes participating in distributed ledger network, and software error in operating DLT system [21]. The solution should meet the following security functional components: FRU_FLT.1 (Degraded fault tolerance), FRU_PRS.1 (Limited priority of service), and FRU_RSA.1 (Maximum quotas) in Common Criteria (CC) [35].

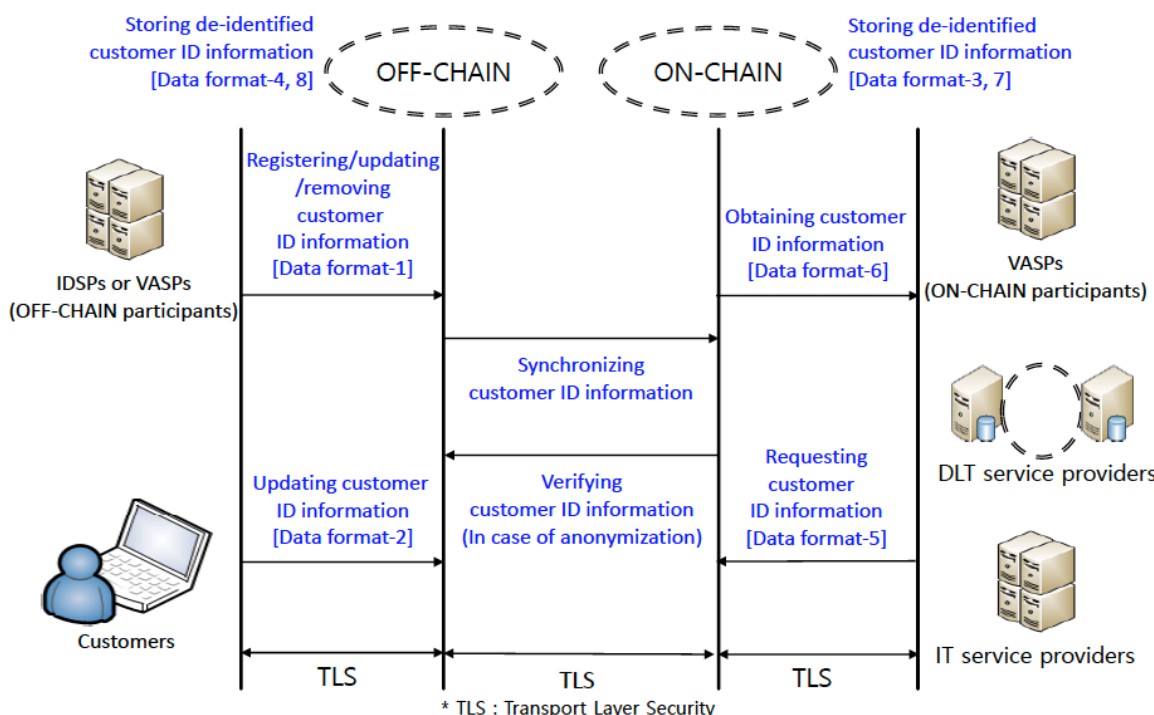

**Figure 6.** Service scenario and data flow.

### 4.2. Service Scenario and Data Flow

Figure 6 describes the relationship between VASP or IDSP and OFF-CHAIN and between customer and OFF-CHAIN. It also describes the relationship between VASP and ON-CHAIN, the relationship between DLT service provider or IT service provider and ON-CHAIN and the relationship between ON-CHAIN and OFF-CHAIN.

The data format used in a customer identification service model based on DLT is Data format-1 to Data format-8, which includes customer's minimal personal data in order to comply with privacy regulations (e.g., GDPR, etc.).

The basic data format for identity information in Table 1 consists of 10 items. In particular, the D-1 to D-4 regarded as personal data will be encrypted with either uni-directional cryptography (e.g., SHA-256, etc.) [36,37] in the ON-CHAIN or bi-directional cryptography (e.g., SEED, AES-128, etc.) [36,37] in the OFF-CHAIN.

In Table 1, the D-0 is a customer number that is a unique number and a hash value for the D-1 to D-3. The D-1 regarded as personal data is a country code that presents the nationality of the customer and is encrypted with uni-directional/bi-directional cryptography in the ON-CHAIN/OFF-CHAIN. The D-2 regarded as personal data is a name that presents the full name of customer who is a natural or legal person and is encrypted with uni-directional/bi-directional cryptography in the ON-CHAIN/OFF-CHAIN. The D-3 regarded as personal data is a certificate that presents a digital certificate of the customer in X.509 [38], which is an international standard of ITU-T and is encrypted with uni-directional/bi-directional cryptography in the ON-CHAIN/OFF-CHAIN. D-4, regarded as personal data, is a digital wallet address that presents a digital wallet address for virtual asset transfers and is encrypted with uni-directional/bi-directional cryptography in the ON-CHAIN/OFF-CHAIN. D-5 is an identifier for VASP, who manages digital wallets for a customer. D-6 is an identifier for the IDSP who verified the customer identity. D-7 presents the validity of the customer identity. As a value of validity, "Valid" means "possible to transact", "Invalid" means "impossible to transact", and "N/A" means "impossible to verify". The VASP should transfer VA only if both the validity of the originator and the validity of the beneficiary are "Valid", which means "possible to transact". D-8 and D-9 are reserved for future use.

**Table 1.** Basic data format.

| Item | Encryption | Value | Description |
|------|-----------|-------|-------------|
| D-0 | - | Customer number | - Unique number given to a customer<br>- Hash Value for D-1 to D-3 |
| D-1 | Encryption | Country code | - Nationality of the customer<br>- e.g., KR (Korea), US (United States), etc. |
| D-2 | Encryption | Name | - Full name of the customer who is natural person or legal person |
| D-3 | Encryption | Certificate | - Certificate of the customer (e.g., X.509) |
| D-4 | Encryption | Digital wallet address | - Digital wallet address for transferring virtual assets |
| D-5 | - | VASP | - Identifier of the virtual asset service provider that manages the customer digital wallet<br>- e.g., Coinbase |
| D-6 | - | IDSP | - Identifier of the service provider who verified the customer identity<br>- It can be same as VASP |
| D-7 | - | Validity | - Validity of the customer identity<br>- e.g., Valid (possible to transact), Invalid (impossible to transact), N/A (impossible to verify) |
| D-8 | - | RESERVED | RESERVED |
| D-9 | - | RESERVED | RESERVED |

In Figure 6, the IDSPs and VASPs as participants in the OFF-CHAIN register, update, and remove customer identity information with Data format-1 (see Table 2) in the OFF-CHAIN. Data format-1 in Table 2 is derived from the basic data format in Table 1.

**Table 2.** Data format-1.

| Item | Necessary/Optional | Value | Description |
|------|-------------------|-------|-------------|
| D-0 | - | RESERVED | RESERVED |
| D-1 | Necessary | Country code | - Nationality of the customer<br>- e.g., KR (Korea), US (United States), etc. |
| D-2 | Necessary | Name | - Full name of the customer who is natural person or legal person |
| D-3 | Necessary | Certificate | - Certificate of the customer (e.g., X.509) |
| D-4 | Necessary | Digital wallet address | - Digital wallet address for transferring virtual assets |
| D-5 | Necessary | VASP | - Identifier of the virtual asset service provider that manages customer digital wallet<br>- e.g., Coinbase |
| D-6 | Necessary | IDSP | - Identifier of the service provider who verified the customer identity<br>- It can be same as VASP |
| D-7 | Necessary | Validity | - Validity of the customer identity<br>- e.g., Valid (possible to transact), Invalid (impossible to transact), N/A (impossible to verify) |

In Figure 6, the customers update their own identity information in the OFF-CHAIN with Data format-2 (see Table 3). Data format-2 in Table 3 is derived from the basic data format in Table 1.

<p style="text-align: center;">**Table 3.** Data format-2.</p>

| Item | Necessary/Optional | Value | Description |
|------|--------------------|-------|-------------|
| D-0 | - | RESERVED | RESERVED |
| D-1 | Necessary | Country code | - Nationality of the customer<br>- e.g., KR (Korea), US (United States), etc. |
| D-2 | Necessary | Name | - Full name of the customer who is natural person or legal person |
| D-3 | Necessary | Certificate | - Certificate of the customer (e.g., X.509) |
| D-4 | Necessary | Digital wallet address | - Digital wallet address for transferring virtual assets |
| D-5 | - | RESERVED | RESERVED |
| D-6 | - | RESERVED | RESERVED |
| D-7 | - | RESERVED | RESERVED |

<p style="text-align: center;">(Note: The customer can update only own digital wallet address as the D-4.).</p>

In Figure 6, the ON-CHAIN stores de-identified customer identity information as Data format-3 (see Table 4). Data format-3 in Table 4 is derived from the basic data format in Table 1.

<p style="text-align: center;">**Table 4.** Data format-3.</p>

| Item | Encryption | Value | Description |
|------|-----------|-------|-------------|
| D-0 | - | Customer number | - Unique number given to a customer |
| D-1 | Uni-direction | Country code | - Nationality of the customer<br>- e.g., KR (Korea), US (United States), etc. |
| D-2 | Uni-direction | Name | - Full name of the customer who is a natural person or legal person |
| D-3 | - | RESERVED | RESERVED |
| D-4 | Uni-direction | Digital wallet address | - Digital wallet address for transferring virtual assets |
| D-5 | - | VASP | - Identifier of the virtual asset service provider that manages the customer digital wallet<br>- e.g., Coinbase |
| D-6 | - | RESERVED | RESERVED |
| D-7 | - | Validity | - Validity of the customer identity<br>- e.g., Valid (possible to transact), Invalid (impossible to transact), N/A (impossible to verify) |

In Figure 6, the OFF-CHAIN stores de-identified customer identity information as Data format-4 (see Table 5). The OFF-CHAIN synchronizes customer identity information to the ON-CHAIN. If customer identity information is anonymized, the ON-CHAIN verifies it from the OFF-CHAIN. Data format-4 in Table 5 is derived from the basic data format in Table 1.

<p style="text-align: center;">**Table 5.** Data format-4.</p>

| Item | Encryption | Value | Description |
|------|-----------|-------|-------------|
| D-0 | - | Customer number | - Unique number given to a customer |
| D-1 | Bi-direction | Country code | - Nationality of the customer<br>- e.g., KR (Korea), US (United States), etc. |
| D-2 | Bi-direction | Name | - Full name of the customer who is natural person or legal person |
| D-3 | Bi-direction | Certificate | - Certificate of the customer (e.g., X.509) |
| D-4 | Bi-direction | Digital wallet address | - Digital wallet address for transferring virtual assets |

**Table 5.** *Cont.*

| Item | Encryption | Value | Description |
|---|---|---|---|
| D-5 | - | VASP | - Identifier of the virtual asset service provider that manages the customer digital wallet<br>- e.g., Coinbase |
| D-6 | - | IDSP | - Identifier of the service provider who verified the customer identity<br>- It can be same as VASP |
| D-7 | - | Validity | - Validity of the customer identity<br>- e.g., Valid (possible to transact), Invalid (impossible to transact), N/A (impossible to verify) |

In Figure 6, the DLT service providers and IT service providers request customer identity information to the ON-CHAIN with Data format-5 (see Table 6) and then obtain customer identity information from the ON-CHAIN as Data format-6 (see Table 7). Data format-5 in Table 6 and Data format-6 in Table 7 are derived from the basic data format as Table 1.

**Table 6.** Data format-5.

| Item | Necessary/ Optional | Value | Description |
|---|---|---|---|
| D-0 | Optional | Customer number | - Unique number given to a customer |
| D-1 | Necessary | Country code | - Nationality of the customer<br>- e.g., KR (Korea), US (United States), etc. |
| D-2 | Necessary | Name | - Full name of the customer who is natural person or legal person |
| D-3 | - | RESERVED | RESERVED |
| D-4 | Necessary | Digital wallet address | - Digital wallet address for transferring virtual assets |
| D-5 | Optional | VASP | - Identifier of the virtual asset service provider that manages the customer digital wallet<br>- e.g., Coinbase |
| D-6 | - | RESERVED | RESERVED |
| D-7 | - | RESERVED | RESERVED |

**Table 7.** Data format-6.

| Item | Necessary/ Optional | Value | Description |
|---|---|---|---|
| D-0 | Optional | Customer number | - Unique number given to a customer |
| D-1 | Necessary | Country code | - Nationality of the customer<br>- e.g., KR (Korea), US (United States), etc. |
| D-2 | Necessary | Name | - Full name of the customer who is natural person or legal person |
| D-3 | - | RESERVED | RESERVED |
| D-4 | Necessary | Digital wallet address | - Digital wallet address for transferring virtual assets |
| D-5 | Optional | VASP | - Identifier of the virtual asset service provider that manages the customer digital wallet<br>- e.g., Coinbase |
| D-6 | - | RESERVED | RESERVED |
| D-7 | Necessary | Validity | - Validity of the customer identity<br>- e.g., Valid (possible to transact), Invalid (impossible to transact), N/A (impossible to verify) |

In Figure 6, the VASPs obtain customer identity information from the ON-CHAIN as Data format-6 (see Table 7). The VASP should transfer VA only if the validity of the originator and the validity of the beneficiary are "Valid", which means "possible to transact". The ON-CHAIN stores de-identified customer identity information as Data format-7 (see Table 8). Data format-7 in Table 8 is derived from the basic data format as Table 1.

**Table 8.** Data format-7.

| Item | Encryption | Value | Description |
|------|-----------|-------|-------------|
| D-0 | - | Customer number | - Unique number given to a customer |
| D-1 | Uni-direction | Country code | - Nationality of the customer<br>- e.g., KR (Korea), US (United States), etc. |
| D-2 | Uni-direction | Name | - Full name of the customer who is natural person or legal person |
| D-3 | - | RESERVED | RESERVED |
| D-4 | Uni-direction | Digital wallet address | - Digital wallet address for transferring virtual assets |
| D-5 | - | VASP | - Identifier of the virtual asset service provider that manages the customer digital wallet<br>- e.g., Coinbase |
| D-6 | - | RESERVED | RESERVED |
| D-7 | - | Validity | - Validity of the customer identity<br>- Set up N/A (impossible to verify) |

In Figure 6, the OFF-CHAIN stores de-identified customer identity information as Data format-8 (see Table 9). Data format-8 in Table 9 is derived from the basic data format as Table 1.

**Table 9.** Data format-8.

| Item | Encryption | Value | Description |
|------|-----------|-------|-------------|
| D-0 | - | Customer number | - Unique number given to a customer |
| D-1 | Bi-direction | RESERVED | RESERVED |
| D-2 | Bi-direction | RESERVED | RESERVED |
| D-3 | Bi-direction | RESERVED | RESERVED |
| D-4 | Bi-direction | RESERVED | RESERVED |
| D-5 | - | VASP | - Identifier of the virtual asset service provider that manages the customer digital wallet<br>- e.g., Coinbase |
| D-6 | - | IDSP | - Identifier of the service provider who authenticated the customer<br>- It can be the same as VASP |
| D-7 | - | Validity | - Validity of the customer identity<br>- Set up N/A (impossible to verify) |

(Note: It should remove permanently the encryption key that encrypts D-1, D-2, D-3, and D-4.).

The way to delete personal data such as D-1 to D-4 stored in OFF-CHAIN is to completely delete the encryption key that encrypted D-1 to D-4. Personal data such as the D-1, D-2, and D-4 stored in ON-CHAIN is encrypted with uni-directional cryptography (e.g., SHA-256, etc.) [36,37] and is very difficult to re-identify.

The Data format-1 (see Table 2) to Data format-8 (see Table 9) mentioned in Figure 6 are described above. Otherwise, when VASPs try to transfer VA over a certain amount (e.g., USD/EUR 1000 [9]) (see Section 3.2); if necessary, the beneficiary VASP could request and obtain the originator's personal data (e.g., address, national identity number, and date and

place of birth) directly with standard protocols (e.g., SAML [5], OpenID Connect [6], and TLS [7,8]) from the originator VASP identified by the ON-CHAIN.

For example, if the VA amount to be transferred is more than 1000 USD/EUR, the process of VA transfer is as follows:

- (Step 1): the originator submits VA transfer information (e.g., originator's name, originator's digital wallet address, beneficiary's name, beneficiary's digital wallet address, and VA amount) to originator-VASP;
- (Step 2): the originator-VASP verifies both the originator's identity information and beneficiary's identity information including validity of their digital wallets by comparing the VA transfer information with customer identity information as Data format-6 (see Table 7) from the ON-CHAIN in the proposed service model;
- (Step 3): the originator-VASP identifies and authenticates the beneficiary-VASP by comparing the VA transfer information with customer identity information as Data format-6 (see Table 7) from the ON-CHAIN in the proposed service model;
- (Step 4): if step 3 is successful, the originator-VASP directly sends the VA transfer information submitted by the originator to beneficiary-VASP using standard protocols (e.g., SAML [5], OpenID Connect [6], TLS [7,8], etc.);
- (Step 5): if step 2, step 3, and step 4 are successful, the originator-VASP transfers VAs to the beneficiary-VASP using existing blockchain;
- (step 6): if step 5 is successful, the originator-VASP directly sends the originator's personal data (e.g., address, national identity number, and date and place of birth) required by the FATF guidance to the beneficiary-VASP using standard protocols (e.g., SAML [5], OpenID Connect [6], TLS [7,8], etc.);
- (Step 7): if step 5 is successful, the beneficiary-VASP identifies and authenticates the originator-VASP by comparing the VA transfer information with customer identity information such as Data format-6 (see Table 7) from the ON-CHAIN in the proposed service model;
- (Step 8): if step 7 is successful, the beneficiary-VASP directly sends the beneficiary's personal data (e.g., address, national identity number, and date and place of birth) required by the FATF guidance to the originator-VASP using standard protocols (e.g., SAML [5], OpenID Connect [6], and TLS [7,8]).

## 5. Security Threats and Requirements

Security threats (STs) to the proposed service model are identified and security requirements (SRs) against the security threats are provided in this section.

There will always be the risk that parties could still use forged identities or their own trusted accomplices to create identities that would not be flagged by any IT network solution. To address this risk, first of all, the IDSP (or VASP acting as IDSP) should consider mitigating identity proofing and enrollment risks in accordance with Article 117–119 of Section IV in [11], which provide the description of impersonation risks and synthetic IDs (involving cyberattacks, data protection and/or security breaches) and identity proofing/enrolment risk mitigation strategies.

### 5.1. Security Threats

Security threats to the proposed service model are identified as follows:

- (ST-1): Malicious customers can register stolen identity information in the OFF-CHAIN. (ST-1) can be misused for money laundering;
- (ST-2): Malicious customers can update their own identity information on the OFF-CHAIN. (ST-2) can be misused for money laundering;
- (ST-3): Malicious IDSPs (or VASPs acting as IDSPs) can register stolen identity information in the OFF-CHAIN. (ST-3) can be misused for money laundering;
- (ST-4): Unsafe cryptographic algorithms can be used for de-identification of customer's personal data stored in the ON-CHAIN and OFF-CHAIN. The (ST-4) can be misused for re-identification of customer's personal data;

- (ST-5): Unsafe cryptographic algorithms can be used for de-identification of customer's personal data during transmission from an entity (i.e., VASP, IDSP, customer) to the OFF-CHAIN. (ST-5) can be misused for re-identification of customer's personal data;
- (ST-6): Unsafe cryptographic algorithms can be used for de-identification of customer's personal data during transmission from the OFF-CHAIN to ON-CHAIN. The (ST-6): Can be misused for re-identification of customer's personal data;
- (ST-7): Unsafe cryptographic algorithms can be used for de-identification of customer's personal data during transmission from the ON-CHAIN to an entity (i.e., IDSP, DLT service provider, IT service provider). (ST-7) can be misused for re-identification of customer's personal data;
- (ST-8): Malicious customers can repudiate an update of their own identity information on the OFF-CHAIN. (ST-8) can be misused for money laundering;
- (ST-9): Malicious IDSPs (or VASPs acting as IDSPs) can repudiate registration of customer's identity information on the OFF-CHAIN. (ST-9) can be misused for money laundering.

The security threats identified above are specific to the proposed service model, not the general IT services.

### 5.2. Security Requirements

Security requirements against the security threats identified in Section 5.1 are provided as follows:

- (SR-1): The proposed service model should enable the IDSP (or VASP acting as IDSP) to register customer's identity information in the OFF-CHAIN and not let the customer register their own identity information in the OFF-CHAIN;
- (SR-2): The proposed service model should enable the customer to update minimal of their own identity information on the OFF-CHAIN and then let VASP confirm the updated identity information on the OFF-CHAIN;
- (SR-3): The proposed service model should confirm if VASP holds an official license issued by government authorities [9] (see Section 3.2) before the VASP participates in the OFF-CHAIN;
- (SR-4): The proposed service model should provide a safe cryptographic algorithm (e.g., SHA-256 and AES-128) [36,37] to de-identify customer's personal data stored in the ON-CHAIN and OFF-CHAIN;
- (SR-5): The proposed service model should provide safe cryptographic protocol (e.g., TLS) [7,8] to de-identify customer's personal data during transmission;
- (SR-6): the proposed service model should verify PKI (public key infrastructure) based digital signature [39] of a customer when the customer updates own identity information on the OFF-CHAIN;
- (SR-7): The proposed service model should verify PKI based digital signature of IDSP (or VASP acting as IDSP) when the IDSP (or VASP acting as IDSP) registers customer's identity information in the OFF-CHAIN.

In Table 10, the SR-1 can prevent the ST-1, the SR-2 can prevent the ST-2, the SR-3 can prevent the ST-3, the SR-4 can prevent the ST-4, the SR-5 can prevent the ST-5 to ST-7, the SR-6 can prevent the ST-8, and the SR-7 can prevent the ST-9.

**Table 10.** Relationship between security threats and requirements.

|      | SR-1 | SR-2 | SR-3 | SR-4 | SR-5 | SR-6 | SR-7 |
|------|------|------|------|------|------|------|------|
| ST-1 | O    |      |      |      |      |      |      |
| ST-2 |      | O    |      |      |      |      |      |
| ST-3 |      |      | O    |      |      |      |      |
| ST-4 |      |      |      | O    |      |      |      |

**Table 10.** *Cont.*

|       | SR-1 | SR-2 | SR-3 | SR-4 | SR-5 | SR-6 | SR-7 |
|-------|------|------|------|------|------|------|------|
| ST-5  |      |      |      |      | O    |      |      |
| ST-6  |      |      |      |      | O    |      |      |
| ST-7  |      |      |      |      | O    |      |      |
| ST-8  |      |      |      |      |      | O    |      |
| ST-9  |      |      |      |      |      |      | O    |

(Note: ST = security threat, SR = security requirement).

## 6. Discussion and Conclusions

The transfers using VAs, in particular cross-border transfers, have recently increased, and regulators are seriously concerned about the issue of money laundering related to VAs. The reason is that due to the anonymity of the existing blockchain and DLT platform, it is practically impossible to verify and obtain the identity of the originator and beneficiary when transferring VAs. To address these issues, the international anti-money-laundering organization (FATF) has imposed CDD to VASPs with the Guidance and Recommendation, and the FATF member countries should regulate VASPs in compliance with the Guidance and Recommendation.

This paper proposes a customer identification service model based on DLT to technically solve a problem with customer identification and to facilitate transfers using VAs. The service model consists of customer (originator/beneficiary), VASPs, IDSPs, DLT security platform, etc. The DLT security platform satisfies security functional requirements such as identification and authentication, security audit, communication protection, cryptography control, access control, privacy protection, data protection, and resource availability. In addition, the DLT security platform, consisting of the ON-CHAIN and OFF-CHAIN, pseudonymizes personal data of customers to satisfy legal requirements such as EU GDPR and Korean Personal Data Protection Act. The service model utilizes the ON-CHAIN and OFF-CHAIN to securely share customer identity information between VASPs to solve a problem with customer identification for AML.

There are several advantages of the proposed service model. (1) The proposed system connects the VASPs using DLT and operates totally independently from the existing blockchain platform that transfers VAs, and it also improves the availability of the proposed system and protects personal data. (2) The OFF-CHAIN is used to securely register, update, and remove customer's minimal identity information including personal data by de-identification technology, and it also improves personal data protection. (3) The ON-CHAIN is used to securely share customer's identity information including personal data by de-identification technology among VASPs and it also improves personal data protection. (4) The ON-CHAIN is used to ensure integrity of customer's identity information including personal data by DLT. (5) The proposed service model can comply with regulation on cross-border VA transfer for AML. (6) The proposed service model can comply with a regulation on the pseudonymisation and right to be forgotten for personal data protection.

There are several reasons why the VASPs and their customers would be interested in accepting the proposed service model. (1) The originator and beneficiary can securely provide their identity information including personal data to VASPs for cross-border VAs transfer in accordance with the regulation on AML. (2) The originator-VASP verifies both the originator's identity information and beneficiary's identity information including the validity of their digital wallets using the proposed service model. (3) The originator-VASP identifies and authenticates the beneficiary-VASP using the proposed service model. (4) The beneficiary-VASP identifies and authenticates the originator-VASP using the proposed service model. (5) The VASP can preserve their customer's identity information including personal data using the proposed service model.

The proposed service model has been developed as Korean ICT standard (project number: 2020-0028) by TTA (Telecommunications Technology Association) PG502 and will

be developed as an international standard by ITU-T (International Telecommunication Unit) SG17. The private companies also will implement the proposed service model as a software system by technology transfer in the future.

**Author Contributions:** Conceptualization, K.P.; methodology, K.P.; validation, K.P. and H.-Y.Y.; formal analysis, K.P.; investigation, K.P.; writing—original draft preparation, K.P.; writing—review and editing, K.P. and H.-Y.Y.; supervision, H.-Y.Y.; project administration, H.-Y.Y. Both authors have read and agreed to the published version of the manuscript.

**Funding:** This research received no external funding.

**Institutional Review Board Statement:** Not applicable.

**Informed Consent Statement:** Not applicable.

**Data Availability Statement:** Not applicable.

**Acknowledgments:** This research was implemented as part of the project "Standardization Lab. for Next-generation Cybersecurity" (Project Number: 2021-0-00112) supported by MSIT (the Ministry of Science and ICT) and IITP (Institute of Information & Communications Technology Planning & Evaluation).

**Conflicts of Interest:** The authors declare no conflict of interest.

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
