# Peer review of "Proposal for Customer Identification Service Model Based on Distributed Ledger Technology to Transfer Virtual Assets"

_2504-2289, doi:10.3390/bdcc5030031_

Round 1
Reviewer 1 Report
The transfers using VAs, in particular cross-border transfers, have recently increased, and regulators are seriously concerned about the issue of money laundering related to VAs. The reason is that due to the anonymity of the existing blockchain and DLT platform, it is practically impossible to verify and obtain the identity of the originator and beneficiary when transferring VAs.
In this manuscript, a customer identification service model based on DLT was proposed to technically solve a problem with customer identification and to facilitate the transfers using VAs. Moreover, the service model consists of customers (originator/beneficiary), VASPs, IDSPs, DLT security platforms, etc. The DLT security platform satisfies security functional requirements such as identification and authentication, security audit, communication protection, cryptography control, access control, privacy protection, data protection, and resource availability. In addition, the authors showed that the service model utilizes the ON-CHAIN and OFF-CHAIN to securely share customer identity information between VASPs to solve a problem with customer identification for AML. As far as I known, both the customer identification service model and the security requirements are correct. Thus, I think this is a good paper.
Author Response
We noted. Thank you.

Reviewer 2 Report
1. References should be numbered according to the order of appearance. The authors start with references [9, 10]?!
2. Authors partially do not use reference numbers but full titles, e.g.:
- "According to ‘KYC optimization using distributed ledger technology’, ..."
Please correct it to the numeric presentation of references.
3. Figure 3. seems to be unnecessary.
4. Please clearly define the advantages of your proposal in comparison to the existing ones.
5. Please list the reason why the users and institutions would be interested to accept your proposals.
Reviewer 3 Report
The authors have done a good job of describing the problem. They have done a good job in reviewing existing methods for combating the problem. They have a done a good job of describing the various organizations that would be involved and the definitions of the terms involved in the virtual asset transfer problem.
The solution involves adding more off-chain verification through trusted third parties for identity verification, but existing certificate authorities are also required because parties are required to have X.509 certificates.
I recommend publication because it adds to set of possible solutions addressing the money laundering problem. Ultimately, there is no IT solution to the problem without some physical verification of the identity of the parties. There will always be the risk of parties could still use forged identities or their own trusted accomplices to create identities that would not be flagged by any IT network solution. The authors should address this issue in their manuscript.
Round 2
Reviewer 2 Report
The authors have resolved all raised issues.
Author Response
We noted. Thank you.